# Learning to Solve New sequential decision-making Tasks with In-Context Learning

## Abstract

Training autonomous agents that can generalize to new tasks from a small number of demonstrations is a long-standing problem in machine learning. Recently, transformers have displayed impressive few-shot learning capabilities on a wide range of domains in language and vision. However, the sequential decision-making setting poses additional challenges and has a much lower tolerance for errors since the environment's stochasticity or the agent's wrong actions can lead to unseen (and sometimes unrecoverable) states. In this paper, we use an illustrative example to show that a naive approach to using transformers in sequential decision-making problems does not lead to few-shot learning. We then demonstrate how training on sequences of trajectories with certain distributional properties leads to few-shot learning in new sequential decision-making tasks. We investigate different design choices and find that larger model and dataset sizes, as well as more task diversity, environment stochasticity and trajectory burstiness, all result in better in-context learning of new out-of-distribution tasks. Our work demonstrates that by leveraging large offline pretraining datasets, our model is able to generalize to unseen MiniHack and Procgen tasks via in-context learning, from just a handful of expert demonstrations per task.

## 1 Introduction

For many real-world application domains such as robotics or virtual assistants, collecting large amounts of data for training an agent can be time consuming, expensive, or even dangerous. Hence, the ability to learn new tasks from a handful of demonstrations is crucial for enabling a wider range of use cases. However, current deep learning agents often struggle to learn new tasks from a limited number of demonstrations. Prior work attempts to address this problem using meta-learning (Schmidhuber, 1987; Duan et al., 2016; Finn et al., 2017a; Pong et al., 2022; Mitchell et al., 2021; Beck et al., 2023) but these methods tend to be difficult to use in practice and require more than a handful of demonstrations or extensive fine-tuning. In contrast, large transformers trained on vast amounts of data can learn new tasks from only a few examples without any parameter updates (Brown et al., 2020; Kaplan et al., 2020; Olsson et al., 2022a; Chan et al., 2022a). This emergent phenomenon is called *few-shot or in-context learning (ICL)*, and is achieved by simply conditioning the model's outputs on a context containing a few examples for solving the task (Brown et al., 2020; Ganguli et al., 2022; Wei et al., 2022).

While in-context learning has been observed in multiple domains from language (Brown et al., 2020) to vision (Chan et al., 2022a), it has not yet been extensively studied in sequential decision-making settings. The sequential decision-making problem poses additional challenges that do not appear in the supervised or self-supervised settings.

In this paper, we study the data distributional properties required to enable in-context learning of sequential decision-making tasks. Our key finding is that in contrast to (self-)supervised learning where the context can simply contain a few different examples (or predictions), in sequential decision-making it is crucial for the context to contain full trajectories (or sequences of predictions) to cover the potentially wide range of states the agent may find itself in at deployment. Translating this insight into a dataset construction pipeline, we demonstrate that we can enable few-shot learning of *unseen* tasks on both the MiniHack and Procgen benchmarks. We additionally perform an extensive study of other design choices such as the model size, dataset size, trajectory burstiness,

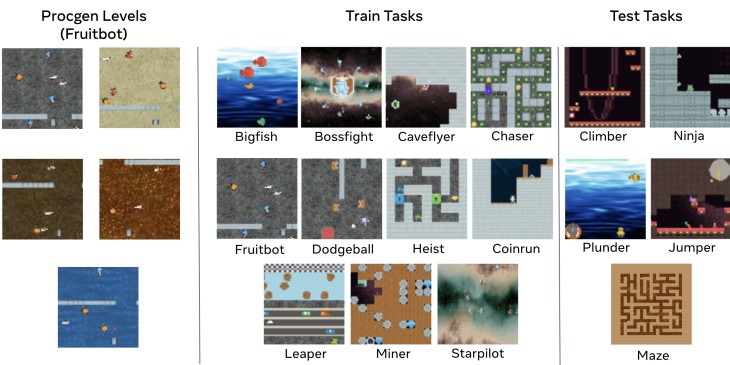

Figure 1: **Illustration of Train and Test Tasks.** (*Left*) A collection of procedurally generated Procgen levels from the `Fruitbot` task, demonstrating the complexity and diversity inherent in the environment's design. (*Middle*) Tasks for training. (*Right*) Tasks for testing. Note that the test tasks are entirely distinct from the training tasks, and each of them is procedurally generated, consisting of multiple levels.

environment stochasticity, and task diversity. Our experiments suggest that larger model and dataset sizes, as well as more trajectory burstiness, environment stochasticity, and task diversity, all lead to better in-context learning when using transformers for sequential decision-making tasks. **Our work is first to show that generalization to new Procgen and MiniHack tasks is possible from just a handful of expert demonstrations and no weight updates using the transformer's in-context learning ability.** This goes beyond the well-studied generalization to new levels which are merely procedurally generated instances of the training task rather than fundamentally new tasks (Cobbe et al., 2020; Igl et al., 2019; Laskin et al., 2020; Raileanu et al., 2020b; Raileanu & Fergus, 2021).

## 2 BACKGROUND

**Markov Decision Processes:** Sequential decision-making tasks can be modeled by Markov Decision Processes (MDPs) (Sutton & Barto, 2018) represented by a tuple $\mathcal{M} = \left( \mathcal{S}, \mathcal{A}, R, P, \gamma, \mu \right)$, where $\mathcal{S}$ is the state space, $\mathcal{A}$ is the action space, $R : \mathcal{S} \times \mathcal{A} \to [R_{\min}, R_{\max}]$ is the reward function , $P : \mathcal{S} \times \mathcal{A} \times \mathcal{S} \to \mathbb{R}_{\geq 0}$ is the transition function, $\gamma \in (0, 1]$ is the discount factor, and $\mu : \mathcal{S} \to \mathbb{R}_{\geq 0}$ is the initial state distribution. We denote the trajectory of an episode by $\tau = (s_0, a_0, r_0, \ldots, s_T, a_T, r_T, s_{T+1})$ where $r_t = R(s_t, a_t)$ and $T$ is the length of the trajectory which can be infinite. If a trajectory is generated by a stochastic policy $\pi : \mathcal{S} \times \mathcal{A} \to \mathbb{R}_{\geq 0}$, $R^\pi = \sum_{t=0}^{T} \gamma^t r_t$ is a random variable that describes the *discounted return* the policy achieves. The objective is to find a policy $\pi^\star$ that maximizes the expected discounted return in the MDP.

**Problem Setting.** In this paper, we are interested in learning new tasks from a small number of demonstrations after pretraining on a large dataset of offline trajectories from different tasks. Formally, we consider a set of tasks $\mathcal{T}$ split into disjoint sets $\mathcal{T}_{train}$ and $\mathcal{T}_{test}$. The train and test tasks have the same action space, but different state spaces, initial state distributions, transition and reward functions. Each task $\mathcal{T}_i \in \mathcal{T}$ is procedurally generated, meaning that it includes $L$ MDPs $\{\mathcal{M}_i^l\}_{l=1}^{L}$ sharing the same state spaces, transition and reward functions, but different initial state distributions. We refer to these as "levels" in this paper as depicted in Figure 1: for MiniHack and Procgen, each level corresponds to a different random seed used to generate the corresponding instance of the task. For each MDP $\mathcal{M}_i$ in our set, we collect $K$ expert demonstrations using reinforcement learning (RL) agents which were pretrained on $\mathcal{M}_i$, to create a dataset $\mathcal{D}_i$ containing trajectories of the form $\tau_i = (s_0^i, a_0^i, \ldots, s_T^i, a_T^i)$, where $T$ is the maximum number of steps in the episode. For the test MDPs $K$ is small, meaning that we only collect a few expert demonstrations for each level of a test task and use them to condition our transformer model.

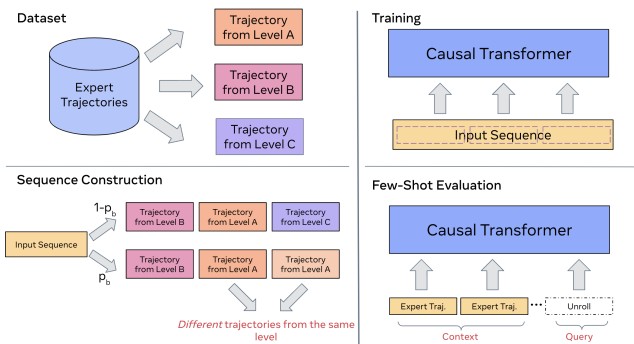

Figure 2: **Experimental Setup:** We create a dataset of expert trajectories by rolling out expert policies on $N$ tasks. Given these expert trajectories, we construct multi-trajectory sequences with trajectory burstiness $p_b$. A sequence is bursty when there are at least two trajectories in the sequence from the same level. However, note that these trajectories are typically different due to the environment's stochasticity. These multi-trajectory sequences then serve as input to the causal transformer, which we train to predict actions. During evaluation, we condition the transformer on an expert trajectory from an unseen task, then rollout the transformer policy until the episode terminates.

## 3 METHODOLOGY

### 3.1 TRAINING DATA

**Tasks:** In order to study ICL for sequential decision-making, we seek domains which are rich, diverse, and of varying complexity. For this reason, we decided to use MiniHack (Samvelyan et al., 2021) and Procgen (Cobbe et al., 2020). MiniHack is a procedurally generated, dungeon-world sandbox based on the NetHack Learning Environment (Küttler et al., 2020) which provides a diverse suite of tasks exhibiting challenges related to navigation, tool use, generalization, exploration and planning. Procgen, similarly to MiniHack, provides 16 procedurally generated game-like tasks to test the generalization abilities of the agent, and additionally features high-dimensional pixel-based observations. In MiniHack and Procgen, each "level" corresponds to a different random seed used to generate the task instance, including the layout, entities and visual aspects.

To generate our expert datasets, we used final checkpoints of E3B (Henaff et al., 2022) policies for MiniHack and PPO (Cobbe et al., 2020) policies for Procgen. Details can be found in B.3.

**Sequence Construction for Transformer Training:** After collecting the expert data, we construct the trajectory sequences for training the transformer. Rather than using single-trajectory sequences as in the Decision Transformer model (Chen et al., 2021) and its successors (Janner et al., 2021a; Lee et al., 2022), we use multi-trajectory sequences (Team et al., 2023a; Laskin et al., 2022). We also remove the explicit conditioning on the episodic return and instead process the sequences in the form of $\{(s_0, a_0, s_1, a_1, r_1, \ldots s_T, a_T)\}$ for each trajectory.

Moreoever, the multi-trajectory sequences adhere to certain distributional properties. First, we extend the concept of burstiness to the sequential decision-making setting. Burstiness was first introduced in Chan et al. (2022a) to describe the quality of a dataset sample where the context contains examples that are similar to the query (for example, from the same class). Here, we extend this idea to trajectory burstiness, which we define as the probability of the context containing atleast two trajectories from the same level . These trajectories can be viewed as the sequential decision-making equivalent of few-shot examples in supervised learning. To construct a bursty trajectory sequence, we sample $E$ trajectories from a given set of $L$ levels in such a way that, with probability $p_b$, there are at least two trajectories coming from the same level. Note that all the trajectories within a sequence come from the same task.

In our main experiments (Section 5), we consider $p_b = 1.0$, meaning that the context contains at least one trajectory from the same level as the query. Lastly, we shuffle the trajectories (and notably not the transitions within these trajectories) to construct a sequence of trajectories that will be used to train the transformer. Figure 2 contains an illustration of our approach.

### 3.2 MODEL TRAINING AND EVALUATION

**Model Architecture and Training Procedure:** For both MiniHack and Procgen, we use standard neural network architectures used in prior work (Samvelyan et al., 2021; Cobbe et al., 2020) to process observations and use a separate embedding layer to process actions. In both cases, we then pass these sequences to a causal transformer (Vaswani et al., 2017b). Finally, we use a cross-entropy loss over all the action predictions in our sequence. Unless otherwise specified, we use a 100M parameter transformer model for MiniHack and a 210M parameter model for Procgen.

**Evaluation:** We evaluate all the pretrained models on *unseen* tasks that differ from those used during training, as shown in Figure 1. For each pretrained model, we conduct two types of evaluations: few-shot and zero-shot. For few-shot evaluations, we condition the model on a handful of expert demonstration (ranging from 1 to 7) and roll out the transformer policy for five episodes. This is repeated for $L$ levels per task, and we aggregate the episodic return across all levels. The evaluation protocol for zero-shot evaluations is identical, with the exception that we do not condition on the expert demonstration. For all our results, we report mean and standard deviation across 3 seeds.

## 4 MOTIVATING EXPERIMENT

Existing state-of-the-art methods that use transformers for decision-making train them on sequences consisting of single trajectories (Chen et al., 2021). In this section, we show that this setup fails to promote in-context learning of new tasks. Instead, we will demonstrate that *training transformers on sequences of multiple trajectories can enable few-shot learning of unseen tasks*. To illustrate this, we compare the performance of a *single-trajectory* transformer trained on single-trajectory sequences, and a *multi-trajectory* transformer trained on multi-trajectory sequences in a simple setting.

The single-trajectory transformer's context consists of all past transitions from the current episode $c_t = \{(s_0, a_0, s_1, a_1, \ldots, s_{t-1}, a_{t-1})\}$, and the transformer has to predict the action given the current state and the context $p(a_t | s_t, c_t)$ for each time-step $t$. Hence, given a new task different from the training ones (meaning that we cannot expect much in-weights (Chen et al., 2022) or zero-shot generalization), the agent can only leverage information from past transitions within the same episode. However, agents typically face different decisions within an episode (*i.e.,* they rarely see the same state twice), making the problem very challenging.

In contrast, a multi-trajectory transformer's context consists of one or more full trajectories from the same task, as well as all past transitions from the current episode. We will consider the one-shot case in this example where $c_t = \{(s_0^0, a_0^0, \ldots, s_T^0, a_T^0), (s_0^1, a_0^1, \ldots, s_{t-1}^1, a_{t-1}^1,)\}$. The transformer has to predict the action for the current state $p(a_t^1 | s_t^1, c_t)$ for all $t$. Hence, when faced with a new task (even if very different from from the training ones), the agent can now leverage information from full trajectories on this task. Note that due to the stochasticity inherent in many environments, the agent will often be faced with new scenarios that don't appear in its context, so simple memorization is not enough and generalization is required. However, since the agent has access to full trajectories, it is less likely that it will encounter states that are too different from the ones in its context so it should perform better.

To verify our hypothesis that *in sequential decision-making, multi-trajectory transformers enable better in-context learning than single-trajectory transformers*, we perform the following experiment. We collect expert demonstrations from 100K levels from the `MiniHack-MultiRoom-N6` task. Using the above protocols, we construct training sequences for the single-trajectory and multi-trajectory transformers and train them using the same hyperparameters[1]. We then evaluate them on the `MiniHack-Labyrinth` task. To ensure a fair comparison, we condition both transformer variants on a single expert demonstration from the test task. **The multi-trajectory transformer with episodic return of $0.780 \pm 0.07$ outperforms the single-trajectory transformer with $-0.323 \pm 0.05$ by a large margin, indicating the importance of training transformers on sequences of trajectories in order to obtain in-context learning of new sequential decision-making tasks (from only a handful of demonstrations and without any weight updates).** For the rest of the results in the paper, we use multi-trajectory transformer as our base model for analysis.

---

[1]We set trajectory burstiness $p_b = 1$ for this experiment.

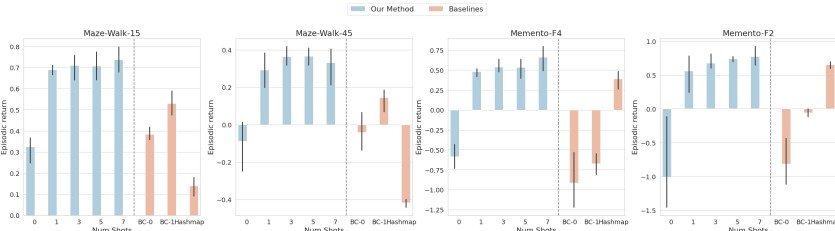

Figure 3: **Performance on New MiniHack Tasks** comparing (1) our multi-trajectory transformer conditioned on different number of demonstrations from the same level, (2) Hashmap baseline conditioned on the same demonstrations, and (3) BC baseline conditioned on zero and one demonstration due to context length constraints. Our model outperforms both baselines when provided with at least one demonstration and its performance improves with the number of demonstration.

## 5 EXPERIMENTAL RESULTS

We next illustrate the in-context learning abilities of the multi-trajectory transformer on unseen MiniHack and Procgen tasks, and compare against single-trajectory and hashmap-based baselines. While many previous works have studied generalization to new levels, to our knowledge we are the first to study generalization to completely different tasks, and we demonstrate promising results.

**Baselines**: We compare against two baselines. **Hashmap (HM):** employs a hashmap over states in order to take the action corresponding to the same state from context (i.e., the expert action). It's important to note that in deterministic environments, this baseline acts as an oracle. However, in stochastic environments, it struggles to generalize to states that are not present in the demonstration. **Behavioural Cloning (BC):** trains a transformer with single trajectory sequences. BC learns a policy to predict the action the expert would take given the history of transitions so far in the current trajectory. During evaluation time, we condition BC on zero (BC-0) and one (BC-1) demonstration. Because of the lack of context, BC struggles to generalize to new tasks.

### 5.1 MAIN RESULTS

**MiniHack** We collect offline data from 12 MiniHack tasks and train a multi-trajectory transformer on sequences comprising eight trajectories from the same level ($p_b = 1$). For evaluation, we test the performance on 4 new MiniHack tasks. We would like to emphasize that environments are stochastic where the stochasticity is induced by sticky actions. The results are reported in Figure 3. Our approach achieves good performance across multiple scenarios, demonstrating that it can learn new tasks with only a handful of demonstrations and without any weight updates. In addition, there is a consistent increase in performance with the addition of more demonstrations across all environments. Notably, our model outperforms all the baselines. When compared to BC, it underscores the significance of training transformers on sequences of trajectories, rather than on single trajectories, to foster in-context learning of new tasks. In contrast to the Hashmap, our model demonstrates the capability to learn a policy that generalizes to unseen states, rather than merely copying actions from its context which is insufficient in this setting.

**Procgen** We also evaluate our approach on a high-dimensional pixel-based domain to test (Figure 1) the generality of our findings. To do so, we collect offline data from 11 Procgen tasks and train a transformer on Procgen sequences compromising of five episodes from the same level ($p_b = 1$). We then evaluate the performance on five unseen tasks. To induce stochasticity, we add gaussian noise to the observations while performing online rollouts[2]. As shown in Figure 4, the performance of our model consistently increases as we condition on more demonstrations while outperforming the baselines. In the case of Maze, the improvement with number of demonstrations is not substantial. We believe this is due to the fact that in Maze, the spatial information of the agent consists of only a few pixels and hence it maybe not be captured by the visual encoder making it difficult to take good actions. In summary, our results show that our method works well in complex pixel environments where learning good state representations is hard, and where states, layouts, dynamics, rewards, colors, textures, backgrounds, and visual features all vary between training and testing.

---

[2]We also evaluate with sticky actions and found no difference in performance (Appendix C.3)

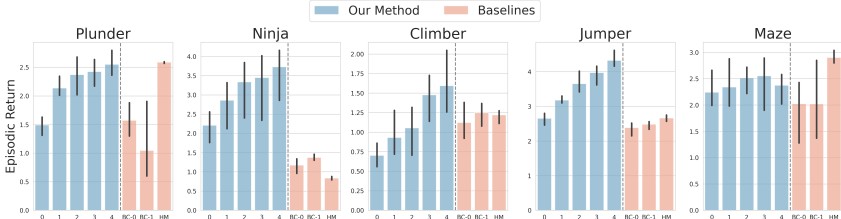

Figure 4: **Performance on New Procgen Tasks** comparing (1) our multi-trajectory transformer conditioned on different number of demonstrations from the same level, (2) Hashmap baseline conditioned on the same demonstrations, and (3) BC baseline conditioned on zero and one demonstration due to context length constraints. Our model outperforms both behavioral cloning baselines, is competitive with Hashmap on Maze and Plunder, and its performance improves with demonstrations.

## 5.2 DETAILED ANALYSIS ON MINIHACK

In this section, we conduct an extensive analysis on the different factors which affect in-context learning for sequential decision-making. We highlight both zero-shot and one-shot results and conclude with a discussion on failure cases. As we've seen in the previous section, adding more demonstrations in the context generally improves the results, so here we focus on the comparison between zero-shot and one-shot which is the most significant.

### 5.2.1 EFFECT OF TRAJECTORY BURSTINESS

First, we analyze the effect of trajectory burstiness. We hypothesize that having demonstrations similar to the query inside the context (*i.e.,* trajectory burstiness) encourages the model to utilize the contextual information. Our results in Figure 6a confirm that in-context learning improves with more trajectory burstiness, resulting in strong one-shot generalization to new tasks for $p_b = 1$. This is consistent with insights from supervised learning Chan et al. (2022a).

### 5.2.2 EFFECT OF ENVIRONMENT STOCHASTICITY

We next investigate the impact of different environment dynamics on in-context learning. For this, we gather offline data using expert E3B policies in both deterministic and stochastic environments. To introduce stochasticity in the environments, we employ sticky actions, a widely utilized technique in deep reinforcement learning (Machado et al., 2018) where previous actions are repeated with some probability ($p = 0.1$ in our experiments).

We gather pretraining datasets from both stochastic and deterministic versions of the `MiniHack-MultiRoom-N6` task, which we use to train two separate variants of the transformer model. We then evaluate each model on unseen environments with either stochastic or deterministic dynamics. Results are shown in a confusion matrix in Figure 5.

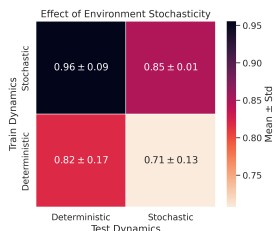

Figure 5: Mean one-shot performance on unseen MiniHack tasks (std. across 3 seeds). Training with stochastic dynamics improves in-context learning.

Models trained with stochastic dynamics exhibit superior performance on unseen tasks, whether their dynamics are deterministic or stochastic. An explanation is that when dynamics are deterministic, the diversity within the training dataset is comparatively low, resulting in identical copies of the trajectory in the context and query. This limits the model's in-context learning ability to pure copying behavior. In contrast, when the training environment dynamics are stochastic, the pretraining dataset exhibits greater diversity and identical copies of the trajectory become rarer, forcing the model to generalize from similar, but not identical, trajectories.

### 5.2.3 EFFECT OF DATASET SIZE, MODEL SIZE AND TASK DIVERSITY

**Dataset Size**: While many studies in NLP (Brown et al., 2020; Kaplan et al., 2020; Olsson et al., 2022a) have reported a positive correlation between dataset size and ICL, few have studied it the

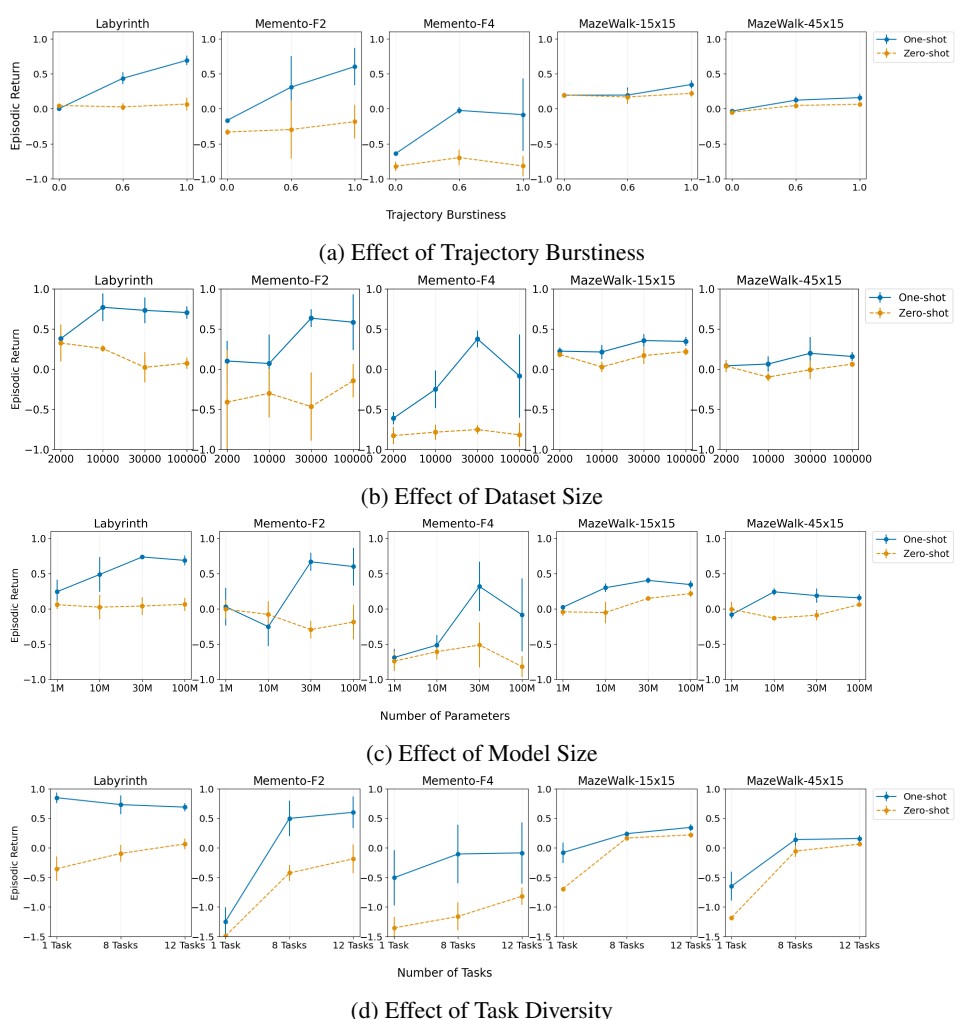

(a) Effect of Trajectory Burstiness

(b) Effect of Dataset Size

(c) Effect of Model Size

(d) Effect of Task Diversity

Figure 6: Mean performance (with std. across 3 seeds) for different levels of trajectory burstiness (a), dataset sizes (b), model sizes (c) and numbers of training tasks (d). These factors all have a positive effect on in-context learning.

realm of sequential decision-making (Team et al., 2023a). Consequently, here we explore the impact of the dataset size on ICL and cross-task generalization. We train models on varying numbers of levels from the same MiniHack task and evaluate them on a different MiniHack task. As seen in Figure 6b, performance on unseen tasks increases with the dataset size and saturates at 30K levels for one-shot models. This is expected as the models improve on their ability to generalize as the dataset increases, even to these new tasks. Moreover, as the dataset sizes increases, we also see the emergence of a performance gap between one-shot and zero-shot evaluations on these new tasks, demonstrating the emergence of in-context learning. We also observe that the performance saturates and subsequently increasing the dataset size above a certain scale (30K levels) yields diminishing returns. We hypothesize that this could be due to insufficient data diversity.

**Model Size**: We next scale the number of model parameters and study how it affects ICL. We evaluate transformer models with 1M, 10M, 30M and 100M parameters, trained on a dataset consisting of 100, 000 levels from 12 different MiniHack tasks. Figure 6c shows that increasing the model size increases ICL on unseen tasks. However, in this case, scaling the model size has diminishing returns and the performance plateaus after 30M parameters.

**Task Diversity**: In this section, we study how ICL behaves as we scale the diversity of tasks used for training. Increasing the diversity of tasks should improve the generalization of the in-weights learning, but its not clear how this should affect a model's ability to 'soft-copy' with ICL. In Fig-

ure 6d we see that above a certain diversity of tasks, ICL ability plateaus, while as the task diversity shrinks, some novel tasks are capable of demonstrating ICL, while others are not. In Section 5.2.4, we investigate the difference between these environments detail.

### 5.2.4 INVESTIGATING FAILURE MODES

In this section, we aim to better understand why our approach performs much better on some tasks than others. To do so, we plot the episodic return and in-context action accuracy for 8 unseen test tasks after training the model on 12 different tasks (see Figure 7). The in-context action accuracy is defined as the percentage of correct actions taken by the agent (*i.e.,* matching the actions provided in the context corresponding to the same states). High in-context action accuracy indicates that the model is able to effectively utilize the demonstration provided in its context. We categorize these results into four categories, each explaining a different learning phenomenon.

**In-Context Learning:** In this category, we have environments like `Labyrinth` and `Memento F2` with both high episodic return and high in-context action accuracy. The high in-context action accuracy indicates that the transformer learns to copy the correct action when it encounters states that appear in the given demonstration, which in turn leads to good performance on the task, as expected.

**In-Weights Learning:** In this category, we have environments like `MazeWalk-15` and `MazeWalk-45` where the episodic return is relatively high but the in-context action accuracy is low. This means that the agent manages to perform well on the new task even without copying actions from its context. This suggests the model is leveraging information stored in its weights during training, also referred to as in-weights learning (Chan et al., 2022a). It is worth noting that `MazeWalk-9` is one of the training environments, which shares similarities with the two test ones. However, note that the test tasks are much harder versions of the training one, so they require some degree of generalization.

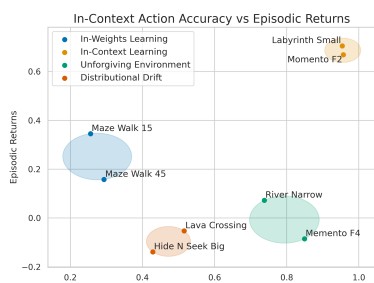

Figure 7: **Investigating Failure Modes:** The figure illustrates different learning phenomena across eight tasks. The x-axis represents the in-context action accuracy, while the y-axis presents the episodic return, with each point in the plot corresponding to an unseen task. We cluster these data points using KMeans clustering and color-code each cluster. Each cluster corresponds to a different learning phenomenon, which are discussed in-depth in Section 5.2.4.

**Unforgiving Environment:** This category includes environments such as `River-Narrow` and `Memento-F4` with high in-context action accuracy but low episodic returns. This indicates that while the agent is able to perform in-context learning and predict the correct action for most states that appear in the demonstration, it still cannot achieve high reward due to reaching unrecoverable states. This is reminiscent of the covariate shift problem in imitation learning, where despite predicting actions well on the expert data, the agent performs poorly at deployment due to small mistakes which put it outside the training distribution (Ross et al., 2011).

**Distributional Drift:** In the last category, we have environments like `Lava-Crossing` and `Hide-N-Seek-Big` where both the episodic return and the in-context action accuracy are low. Here, the agent is unable to perform in-context learning and the test tasks are too different from the training ones for in-weights learning to be effective.

## 6 RELATED WORK

Here we give an overview of related work from the areas of learning from a few demonstrations, using transformers for sequential decision-making, and in-context learning with transformers. We provide additional references in Appendix A.

**Learning from a Few Demonstrations:** Several other works aim to learn from only a few demonstrations using them as inverse curricula for solving very hard exploration problems, even if the demonstrations don't contain any actions (Resnick et al., 2018; Salimans & Chen, 2018). However,

none of these works leverage the in-context learning abilities of transformers to achieve few-shot offline learning of new tasks. There is also a large body of work focusing on meta-reinforcement learning, but this requires online interaction and feedback from the environment in order to adapt to new tasks at test time (Schmidhuber, 1987; Duan et al., 2016; Finn et al., 2017a; Pong et al., 2022; Mitchell et al., 2021; Beck et al., 2023), which we do not assume. Instead, our aim is to learn to solve new tasks from a small number of offline demonstrations.

**Transformers for Sequential Decision-Making:** The success of transformers in natural language processing (Vaswani et al., 2017a) has motivated their application to sequential decision-making (Raileanu et al., 2020a; Janner et al., 2021b; Reid et al., 2022; Ajay et al., 2022; Correia & Alexandre, 2022; Yamagata et al., 2022; Melo, 2022; Reed et al., 2022; Zheng et al., 2022; Chen et al., 2022; Lu et al., 2023; Ryoo et al., 2022; Lin et al.; Sun et al., 2023; Hu et al., 2023; Xu et al., 2023; Sudhakaran & Risi, 2023), with several works emphasizing their limitations in this setting (Brandfonbrener et al., 2022; Paster et al., 2022; Siebenborn et al., 2022). The Decision Transformer (DT) (Chen et al., 2021) was one of the first works to treat policy optimization as a sequence modelling problem and train transformer policies conditioned on the episode history and future return. Inspired by DT, Multi-Game Decision Transformer (MGDT) (Lee et al., 2022) trains a transformer to solve multiple Atari games, but the generalization capablities are limited without additional fine-tuning on the unseen tasks. More similar to our work, Prompt-DT (Xu et al., 2022) shows that DTs conditioned on an explicit prompt exhibit few-shot learning of new tasks (*i.e.,* with different reward functions) in the same environment (*i.e.,* with the same states and dynamics). Similarly, Melo (2022) show that transformers are meta-reinforcement learners, but they require access to rewards and multiple demonstrations at test time. While Team et al. (2023b) also demonstrates fast online learning of new tasks, their model is trained on billions of tasks, whereas we consider the few-shot offline learning setting and a task space which is orders of magnitude smaller. In contrast with all these works, our goal is to learn to generalize to *completely new tasks* (*i.e.,* with different states, dynamics, and rewards) from a small number of expert demonstrations. Our work is also first to extensively study how different factors (such as task diversity, trajectory burstiness, environment stochasticity, model and dataset size) influence the emergence of in-context learning in sequential decision-making.

**In-Context Learning with Transformers:** In-context learning (ICL), first coined in (Brown et al., 2020), is a phenomenon where a model learns a completely new task simply by conditioning on a few examples, without the need for fine-tuning or parameter updating. Many works thereafter study this phenomenon (von Oswald et al., 2022; Chan et al., 2022b; Akyürek et al., 2022; Olsson et al., 2022b; Hahn & Goyal, 2023; Xie et al., 2021; Dai et al., 2022). For example, (Chan et al., 2022a) analyze what makes large language models perform well on few-shot learning tasks through the lens of data properties. Their findings suggest that ICL naturally arises when data distribution follows a power law (*i.e.,* Zipfian) distribution and exhibits inherent "burstiness." In Kirsch et al. (2022), the authors demonstrate that ICL emerges as a function of the number of tasks and model size, with a clear phase transition between instance memorization, task memorization, and generalization. More recently, Garg et al. (2022) show that standard transformers can ICL learn entire function classes such as linear functions, sparse linear functions, and two-layer MLPs. While all these works are confined to the supervised learning setting, our work aims to study what drives the emergence of ICL in the sequential decision-making setting.

## 7 CONCLUSION

In this work, we perform an in-depth study of the different factors which influence in-context learning for sequential decision-making tasks. We find that a key ingredient during pretraining is to include entire trajectories in the context, which belong to the same environment level as the query trajectory. In addition, we find that larger model and dataset sizes, as well as more task diversity, environment stochasticity, and trajectory burstiness, all result in better few-shot learning of out-of-distribution tasks. Leveraging these insights, we are able to train models which generalize to unseen tasks at test time on both the MiniHack and Procgen benchmarks, using only a handful of expert demonstrations and no additional weight updates. To our knowledge, we are the first to show cross-task generalization on MiniHack and Procgen. We also probe the limits of our approach by highlighting different failure modes, such as those caused by environment stochasticity and covariate shift, which we believe constitute important directions for future work.

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

## A  ADDITIONAL RELATED WORK

In many application domains such as robotics, collecting data is expensive and potentially dangerous (Schulman et al., 2016; Englert & Toussaint, 2018). Hence, learning new tasks from single demonstrations is an important problem in sequential decision-making due to its practical appeal (Atkeson & Schaal, 1997; Mohseni-Kabir et al., 2015; Duan et al., 2017; Finn et al., 2017b; Mandi et al., 2022). However, existing algorithms require large-scale datasets to learn new skills (Jang et al., 2022). Hence, we would like to be able to train on a set of tasks for which we have available data and generalize to new ones given a limited number of demonstrations.

Another related work is Algorithmic Distillation (AD) (Laskin et al., 2022), which aims to learn a policy improvement operator by training a transformer on sequences of trajectories of increasing returns. However, AD assumes access to multiple model checkpoints with different proficiency levels and hasn't demonstrated generalization to entirely new tasks or environments.

## B  EXPERIMENTAL SETUP

### B.1  MODEL DETAILS

**Model:** For most of our MiniHack experiments, we use a 100M causal transformer model with `num_layers = 12`, `d_model = 768`, and `num_heads = 12`. For our model scaling experiments presented in Section 5.2.3, we consider the following values for `num_layers`, `d_model`, and `num_heads` presented in Table 1. For procgen experiments, we use a 210M causal transformer with `num_layer = 12`, `d_model = 1024` and `n_heads = 16`.

| Number of Parameters | num_layers | d_model | num_heads |
|:---:|:---:|:---:|:---:|
| 1M | 4 | 128 | 4 |
| 10M | 6 | 256 | 8 |
| 30M | 8 | 512 | 8 |

Table 1: List of transformer model configurations with varying number of parameters, including details on the number of layers (`num_layers`), model dimension (`d_model`), and the number of attention heads (`num_heads`).

For embedding the minihack observations, we use the `NetHackStateEmbeddingNet` from https://github.com/facebookresearch/e3b/blob/main/minihack/src/models.py. For embedding actions and rewards, we follow the same procedure as Chen et al. (2021). For all our experiments, we stack 3 episodes along the sequence axis and restrict the context size to 900 tokens. Since the episodes are of variable length, we stack the three episodes contiguously and then pad the rest of the sequence with padding tokens.

**Training:** For all our experiments, we used AdamW optimizer (Shen et al., 2023) for optimization with cosine annealing learning rate scheduler. We train all our models for 25 epochs.

### B.1.1  COMPUTE

We use 8 GPUs, each with 32GB of RAM, leveraging PyTorch's Distributed Data Parallel (DDP) capabilities for training. The largest model in our experiments (100M), trained on a dataset comprising 100K MiniHack levels with a batch size of 64, took approximately 42 hours (1 day and 18 hours) for complete training. We emphasize that it is feasible to conduct the training with a single GPU; however, this approach would extend the training duration. For inference tasks, we utilize 1 GPU.

### B.2  ENVIRONMENT DETAILS

### B.2.1  MINIHACK

**Observation Space -** The observation space in MiniHack is designed as a dictionary-style structure. It primarily borrows keys from the foundational NetHack Learning Environment (NLE), with the

addition of a few unique keys specific to MiniHack. To get the necessary observations from the environment, relevant options need to be specified during the initialization process. In this paper, we focus on a few key observation types:

1. **Glyphs**: This refers to a 21x79 matrix comprising glyphs, which are identifiers of entities present on the map. Each glyph is unique and represents a specific entity. They are represented as integers ranging from 0 to 5991.

2. **Blstats**: This is a 26-dimensional vector that displays the status line found at the screen's bottom. It provides information about the player character's location, health status, attributes, and other vital statuses.

3. **Messages**: This is a 256-dimensional vector that represents the UTF-8 encoding of the message displayed on the screen's top.

**Action Space -** In this work, all of our environments are navigation-based, and therefore, the action space consists of eight compass directions. Additionally, we create a padding action represented by the number 31 for padding purposes.

**Rewards -** In all MiniHack environments, the rewards are sparse. In other words, the agent receives a reward of +1 if it successfully solves the task at hand, and 0 otherwise. The environment also penalizes the agent for bumping into lava, monsters, or walls.

In our work, we consider the following environments -

1. `MiniHack-MultiRoom-N6-v0` - In this task, the agent must navigate through six rooms and reach the goal. Upon reaching the goal, the agent receives a reward of $+1$, otherwise, the reward is $0$. Collisions with walls incur a penalty of $-0.01$.

2. `MiniHack-MultiRoom-N6-Lava-v0` - In this task, the agent navigates through six rooms with lava-covered walls. The task is completed with a reward of $+1$ when the goal is reached; otherwise, the reward is $0$. Collisions with lava result in immediate death.

3. `MiniHack-MultiRoom-N4-Lava-v0` - This task is similar to `MiniHack-MultiRoom-N6-Lava-v0`, but with only four rooms. The agent must avoid lava-filled walls to complete the task. The task is completed with a reward of $+1$ when the goal is reached; otherwise, the reward is $0$. Collisions with lava result in immediate death.

4. `MiniHack-MazeWalk-9x9-v0` - The agent navigates a $9 \times 9$ maze. Upon reaching the goal, the agent receives a reward of $+1$ otherwise, the reward is $0$. Collisions with walls incur a penalty of $-0.01$.

5. `MiniHack-CorridorBattle-Dark-v0` - The agent must battle the monsters strategically in a dark room, using corridors for isolation. The agent must keep track of its kill count. A reward of $+1$ is given upon survival, otherwise the agent receives a reward of $0$.

6. `MiniHack-MultiRoom-N6-LavaMonsters-v0` - The agent must navigate through six rooms filled with monsters and lava. To successfully complete this task, the agent must avoid the lava walls and monsters. The agent receives a reward of $+1$, otherwise $0$.

7. `MiniHack-MultiRoom-N6-Open-v0` - The agent navigates through six rooms without doors. The reward structure is identical to that of `MiniHack-MultiRoom-N6-v0`.

8. `MiniHack-MultiRoom-N10-LavaOpen-v0` - In this task, the agent navigates through ten rooms to reach the goal. Collisions with lava walls result in immediate death. Upon reaching the goal, the agent receives a reward of $+1$.

9. `MiniHack-SimpleCrossingS11N5-v0` - Here, the agent reaches the goal on the other side of the room. The agent receives a reward of $+1$ for reaching the goal, a penalty of $-0.01$ for hitting the walls, and $0$ otherwise.

10. `MiniHack-SimpleCrossingS9N2-v0` - This task is an easier version of `MiniHack-SimpleCrossingS11N5-v0`.

11. `MiniHack-River-Narrow-v0` - The agent crosses a river by using boulders to create a dry path, aiming to reach the goal on the other side. The agent receives a reward of $+1$ upon reaching the goal.

12. `MiniHack-HideNSeek-v0` - The agent is placed in a room filled with sight-blocking trees and clouds. The agent must avoid monsters while navigating the environment to reach the goal swiftly.

13. `MiniHack-MultiRoom-N2-Monster-v0` - The agent must navigate through two rooms, avoiding monsters. It receives a reward of +1 upon reaching the goal and otherwise 0.

14. `MiniHack-LavaCrossingS11N5-v0` - This task is similar to SimpleCrossing, but the environment is filled with several lava streams running across the room, either horizontally or vertically. The agent needs to avoid the lava, as touching it would result in immediate death. The agent receives a reward of +1 upon the completion of the task.

15. `MiniHack-Memento-F2-v0` - In this task, the agent is presented with a cue at the beginning of the episode and the agent must navigate through the corridor. The end of corridor leads to a two-path fork and the agent should choose one direction based on the cue it saw in the beginning. Upon choosing the correct path and reaching the goal, the gent receives +1 reward and -1 for stepping on the trap.

16. `MiniHack-Memento-F4-v0` - In this task, the agent is presented with a cue at the beginning of the episode and the agent must navigate through the corridor. The end of corridor leads to a four-path fork and the agent should choose one direction based on the cue it saw in the beginning. Upon choosing the correct path and reaching the goal, the gent receives +1 reward and -1 for stepping on the trap.

17. `MiniHack-MazeWalk-15x15-v0` - This task is a slightly harder task than `MiniHack-MazeWalk-15x15-v0`. The agent navigates a $15 \times 25$ maze. Upon reaching the goal, the agent receives a reward of $+1$ otherwise, the reward is $0$. Collisions with walls incur a penalty of $-0.01$.

18. `MiniHack-MazeWalk-45x15-v0` - This task is a very challenging task than `MiniHack-MazeWalk-9x9-v0` The agent navigates a $45 \times 15$ maze. Upon reaching the goal, the agent receives a reward of $+1$ otherwise, the reward is $0$. Collisions with walls incur a penalty of $-0.01$.

19. `MiniHack-Labyrinth-Small-v0` - This task is a very challenging task than `MiniHack-MazeWalk-9x9-v0` The agent navigates a $45 \times 15$ maze. Upon reaching the goal, the agent receives a reward of $+1$ otherwise, the reward is $0$. Collisions with walls incur a penalty of $-0.01$.

For the task diversity experiments presented in Section 5.2.3, we use the environments in Table 3. Note that we randomly sample these environments to avoid selection bias.

### B.2.2 PROCGEN

1. `Bigfish` – In the task, the agent must only eat fish smaller than itself to survive. It receives a small reward for eating a smaller fish and a large reward for becoming bigger than all other fish. The episode ends once it happens.

2. `Bossfight` – In this task, the agent controls a starship, dodging boss attacks until the boss's shield goes down, then damages the boss to earn small rewards. After repeated damage and rewards, the boss is destroyed and the player wins a large final reward.

3. `Caveflyer` – In this task, the agent controls a starship through a cave network to reach a goal ship, moving like in Asteroids. Most reward comes from reaching the goal, but more can be earned by destroying targets along the way. There are lethal stationary and moving obstacles.

4. `Chaser` – In this task, the agent collects green orbs while avoiding enemies, with stars temporarily making enemies vulnerable. Eating a vulnerable enemy spawns an egg that hatches into a new enemy. The player gets small rewards per orb and a large reward for completing the level.

5. `Climber` – In this task, the agent climbs platforms collecting stars, getting small rewards per star and a big reward for all stars, ending the level. Lethal flying monsters are scattered throughout.

| Number of Tasks | Environments |
|---|---|
| 1 Task | `MiniHack-MultiRoom-N6-v0,` |
| 4 Tasks | `MiniHack-LavaCrossingS11N5-v0` |
| | `MiniHack-MultiRoom-N6-v0` |
| | `MiniHack-River-Narrow-v0` |
| | `MiniHack-MultiRoom-N10-LavaOpen-v0` |
| 8 Tasks | `MiniHack-SimpleCrossingS11N5-v0` |
| | `MiniHack-MultiRoom-N10-LavaOpen-v0` |
| | `MiniHack-MultiRoom-N6-v0` |
| | `MiniHack-MultiRoom-N6-Lava-v0` |
| | `MiniHack-MultiRoom-N4-Lava-v0` |
| | `MiniHack-MazeWalk-9x9-v0` |
| | `MiniHack-MultiRoom-N6-LavaOpen-v0` |
| | `MiniHack-CorridorBattle-Dark-v0` |
| 12 Tasks | `MiniHack-MultiRoom-N6-v0` |
| | `MiniHack-MultiRoom-N6-Lava-v0` |
| | `MiniHack-MazeWalk-9x9-v0` |
| | `MiniHack-CorridorBattle-Dark-v0` |
| | `MiniHack-MultiRoom-N6-LavaMonsters-v0` |
| | `MiniHack-MultiRoom-N6-Open-v0` |
| | `MiniHack-MultiRoom-N10-LavaOpen-v0` |
| | `MiniHack-SimpleCrossingS11N5-v0` |
| | `MiniHack-SimpleCrossingS9N2-v0` |
| | `MiniHack-River-Narrow-v0` |
| | `MiniHack-HideNSeek-v0` |
| | `MiniHack-MultiRoom-N2-Monster-v0` |

Table 2: This table illustrates the different environments used for the pretraining.

6. `Coinrun` – In this task, the player must collect a coin on the far right, starting on the far left, dodging saws, pacing enemies, and lethal gaps. Velocity info is no longer included, increasing the difficulty versus prior versions.

7. `Dodgeball` – In this task, the agent navigates a room with walls and enemies, losing if they touch a wall. The player and enemies move slowly, with enemies throwing balls and the player throwing balls in their facing direction. Hitting all enemies unlocks the exit platform; exiting earns a large reward.

8. `Fruitbot` – In this task, the agent controls a robot collecting fruit and avoiding non-fruit objects in a scrolling gap game, getting rewards or penalties. Reaching the end brings a large reward.

9. `Heist` – In this task, the agent controls a character in a maze filled with monsters. The goal is to reach the exit while collecting coins and avoiding monster attacks. More coins collected and reaching the exit results in a higher reward.

10. `Jumper` – In this task, the goal is to navigate the world to find a carrot, using double jumps to reach tricky platforms and avoid deadly spikes. A compass shows direction and distance to the carrot. The only reward comes from collecting the carrot which ends the episode.

11. `Leaper` – In this task, the agent crosses lanes of moving cars then hops logs on a river to reach the finish line and get a reward. Falling in the river ends the episode.

12. `Miner` – In this task, the agent digs through dirt to collect all diamonds in a world with gravity and physics, avoiding deadly falling boulders. Small rewards are given for diamonds, and a large reward for completing the level by exiting after collecting all diamonds.

13. `Maze` – In this task, the agent must navigate to find the sole piece of cheese. Upon reaching the cheese, the agent receives a large reward.

14. `Ninja` – As a ninja, the agent jumps across ledges avoiding bombs, charging jumps over time, and can clear bombs by tossing throwing stars. The player is rewarded for collecting the mushroom that ends the level.

15. `Plunder` - In this task, the player controls a ship firing cannonballs at enemy pirate ships before the on-screen timer runs out, while avoiding friendly ships, with rewards for hits and penalties for misses or hitting friendlies. Firing advances the timer, so ammunition must be conserved. Wooden obstacles can block line of sight to enemies.

16. `Starpilot` - A side scrolling shooter where all enemies target and fire at the player, requiring quick dodging to survive, with varying enemy speeds and health. Clouds block vision and meteors block movement.

| Train Tasks | Test Tasks |
|---|---|
| Bigfish, Bossfight, Caveflyer, Chaser | Climber, Ninja |
| Fruitbot, Dodgeball, Heist, Coinrun | Plunder, Jumper |
| Leaper, Miner, Starpilot | Maze |

Table 3: This table illustrates the Train and Test split of Procgen

Note that in some environments, the episodes are very long, making it harder to fit the full context into the transformer. To address this, we truncate episodes to 200 steps during both training and testing. In test environments, most games have fewer than 200 episode steps, except for Plunder where we only show the first 200 timesteps.

### B.3 DATASET DETAILS

**Data Collection for training -** As mentioned in Section 3.1, we use E3B policies (Henaff et al., 2022) for our data collection. We first train E3B policies following the protocol mentioned in Henaff et al. (2022) on all the above-mentioned environments individually until the performance is optimal. We then take these expert E3B models and use them for data collection. For each level in the environment, we rollout the expert E3B policies for five episodes and collect the observations (`glpyhs, messages, blstats`), actions and reward information. We repeat this for $L$ levels per environment. Additionally, we filter out the bad trajectories which did not solve the task and only utilize the optimal trajectories.

For Procgen, we train a PPO policy for 25M timesteps until convergence on 10K levels for each task. We then use the final checkpoints to collect the trajectories.

**Evaluation Data -** For evaluation on unseen environments (`Labyrinth, Memento-F2, Memento-F4, MazeWalk 15x15` and `MazeWalk 45x19`), we use human expert trajectories instead of E3B policies. The main reason for this is that the E3B policies are incentivized to explore more, which might hinder the agent from solving the task optimally. For each evaluation environment, we manually collect trajectories on 20 levels and use these human expert trajectories as prompts for evaluation.

For Procgen, we use PPO trajectories as expert demonstrations. For each evaluation, we consider test 100 levels and rollout 5 episodes per level and aggregate the returns.

**Multi-Task Data** - For all the results presented in Section 5.2 we use the offline data collected from 12 tasks for pretraining. We consider a total number of levels $L = 100,000$ across all tasks or $8,300$ levels per task. For the task diversity experiments, we consider $100,000, 25000, 12,500,$ and $8,300$ levels per task for 1, 4, 8, and 12 tasks, respectively.

Similarly for Procgen results, we use the offline data collect from 11 tasks for pretraining and for each task, we collect trajectories from 10k levels.

## C ADDITIONAL RESULTS

### C.1 IMPACT OF REWARD TOKENS

In this section, we study the impact of reward tokens in the context on ICL. Since we only use expert data for our training, we hypothesize that the reward tokens do not play any cruicial role in the learning process. Towards this end, we perform experiments where we remove reward tokens

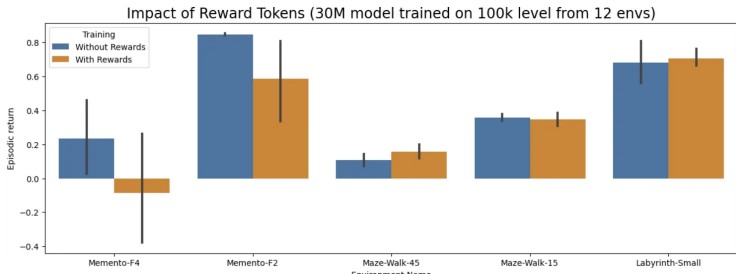

Figure 8: **Impact of Reward Tokens**: This figure illustrates the impact of reward tokens on the performance. Overall, reward tokens do not seem to play a crucial role in the performance across on all five MiniHack tasks.

from the context and keeping just state and action tokens. We contrast the performance with the model trained with reward tokens. As we can see from

## C.2 TRAINING RESULTS

In this section, we report the performance of our pre-trained models on the training environments. We adhere to the same evaluation protocol specified in Section 3.2 and report the mean and standard deviation of the episodic return across three model seeds.

### C.2.1 MINIHACK

We present the results in Figure 9, 10, and 11. Overall, the performance is good on most of the environments and there is no big difference between the zero-shot and one-shot performance. This means that the model is utilizing the in-weights information to solve these tasks and conditioning on one-shot demonstration is not helping much. However, we notice that the model struggles to perform well on some tasks like /textttRiver Narrow, Corridor Battle and `Lava Crossing`. We speculate that these environments are hard to solve and there is a high penalty for death.

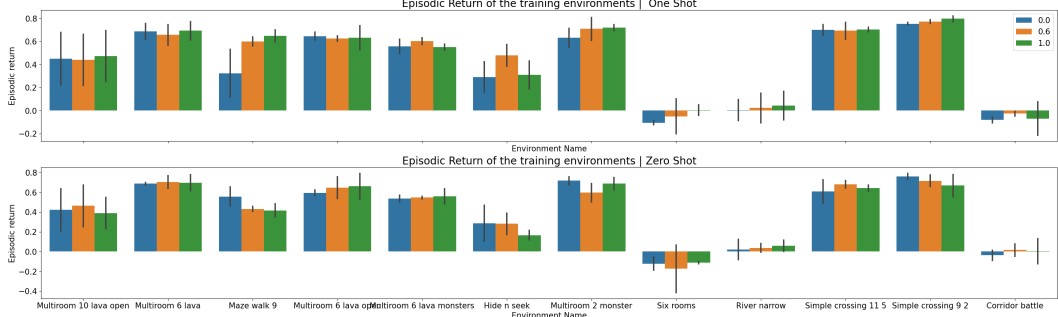

Figure 9: **Performance on Test Levels from 12 Train Tasks** - This figure illustrates the episodic returns on 12 environments used for pre-training, across three different trajectory burstiness values; $p_b \in \{0.0, 0.6, 1.0\}$. Overall, the performance is good on most of the environments and there is no big difference between the zero-shot and one-shot performance. This means that the model is utilizing the in-weights information to solve these tasks, as expected. However, we notice that the agent's performance on `Corridor Battle`, `Six Rooms`, `River Narrow` is rather poor.

### C.2.2 PROCGEN

We present results in Figure 12. We can see that the transformer model is able to perform consistently well across all the training environments. We also notice that there is not big difference in zero-shot and few-shot performance.

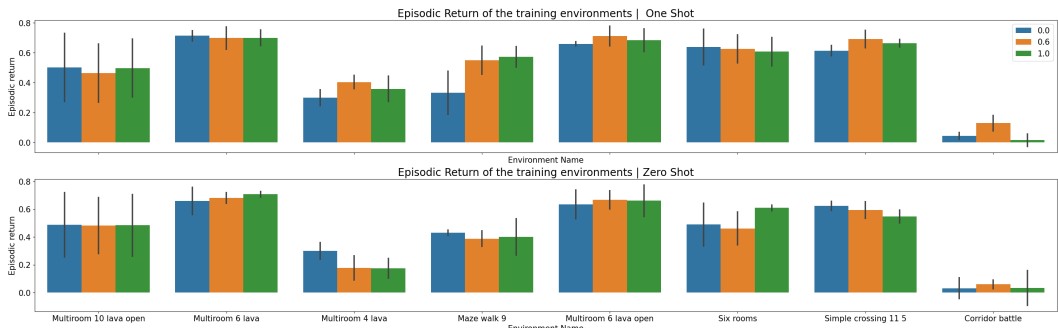

Figure 10: **Performance on Test Levels from 8 Train Tasks** - This figure illustrates the episodic returns on 8 pretraining environments across three different trajectory burstiness values; $p_b \in \{0.0, 0.6, 1.0\}$. Overall, the performance is good on most of the environments and there is no big difference between the zero-shot and one-shot performance. This means that the model is utilizing the in-weights information to solve these tasks, as expected. However, we notice that the performance on `Corridor Battle` is rather poor.

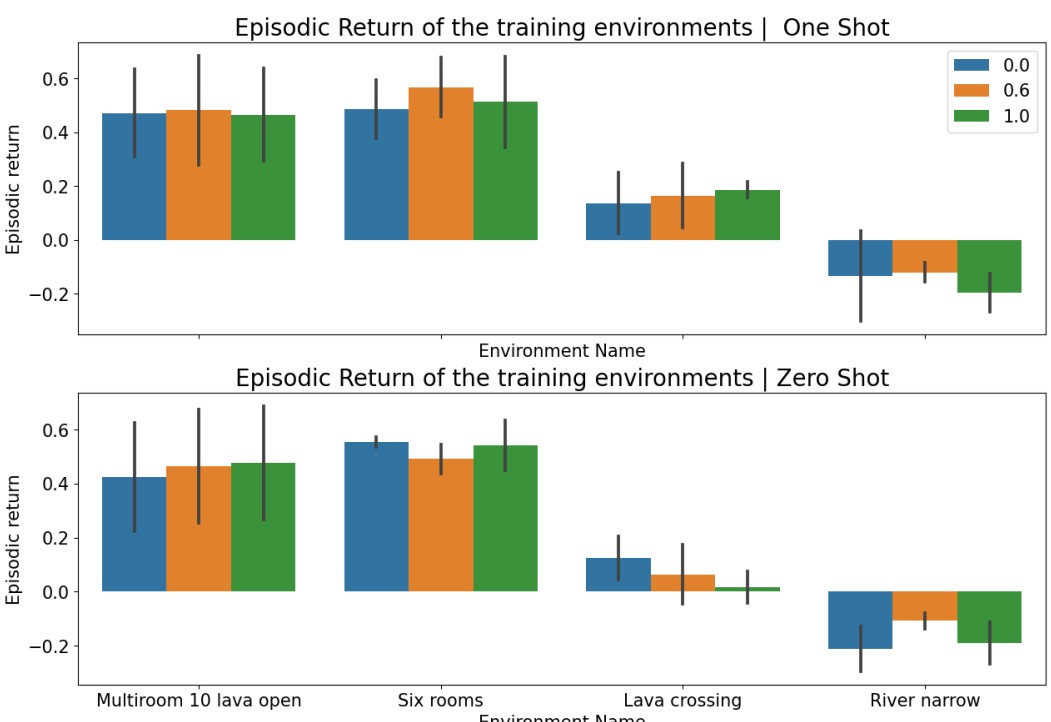

Figure 11: **Performance on Test Levels from 4 Train Tasks** - This figure illustrates the episodic returns on 4 pretraining environments across three different trajectory burstiness values; $p_b \in \{0.0, 0.6, 1.0\}$. Overall, the performance is good on most of the environments and there is no big difference between the zero-shot and one-shot performance. This means that the model is utilizing the in-weights information to solve these tasks, as expected. However, we notice that the performance on `Lava Crossing, River Narrow` is rather poor.

### C.3 PROCGEN STICKY ACTION RESULTS

In this section, we perform evaluations on Procgen test environments with sticky actions. We follow similar evaluation protocol as mentioned in Section 5 and report the performance across 5 procgen tasks. As we can see, the performance of our model is consistent with the performance in Figure 13. While we outperform the most of the baselines, the Hashmap is competitive in case of climber and

| Hyperparameters | Values |
|---|---|
| Batch Size | 64 |
| Number of Levels | { 10000, 30000, 100000} |
| Context Size | 900 |
| Vocabulary Size | 32 |
| Trajectory Burstiness $p_b$ | { 0.0, 0.6, 1.0 } |
| Sticky Probability | 0.2 |
| Number of Epochs | 25 |
| Learning Rate | 0.0001 |
| Learning Rate Scheduler | Consine Annealing |
| Minimum Learning Rate | 1.0e-06 |
| Model Size | {128, 256, 512, 768, 1024 } |
| Number of Layers | { 4, 6, 8, 12,} |
| Number of Heads | { 4, 8, 12 } |
| Vocabulary Size | 32 |
| Dropout | 0.2 |
| Nethack Hidden Size | 64 |
| Nethack Embedding Size | {128, 256, 512, 768 } |
| Crop Size | 12 |
| Nethack Number of Layers | 2 |
| Nethack Dropout | 0.1 |

Table 4: List of hyperparameters used in our experiments

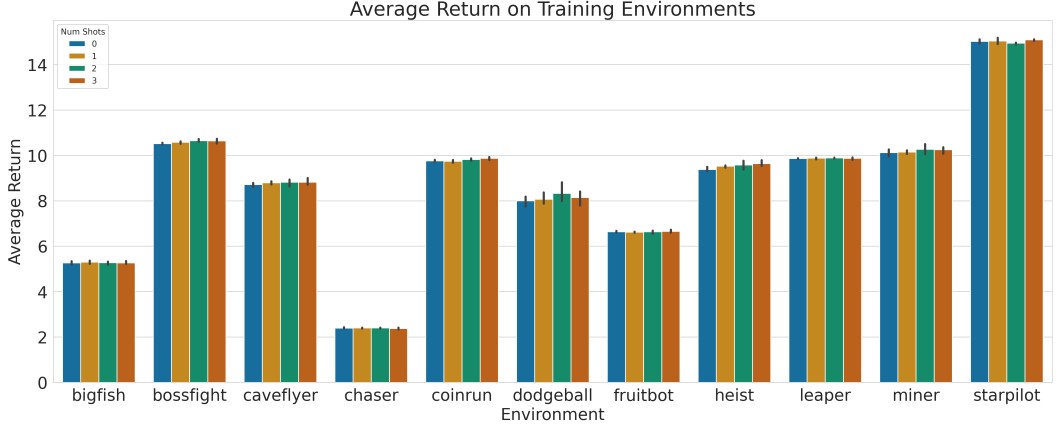

Figure 12: Procgen Train Environments Results. We observe that the model is able to perform well across all the training environments.

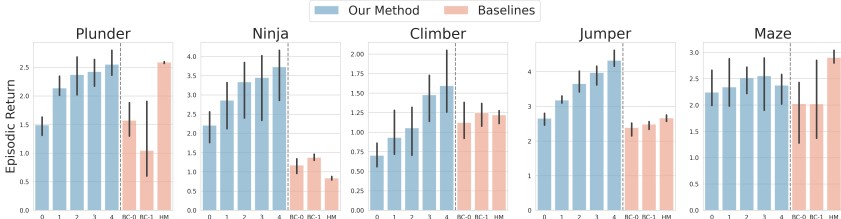

Figure 13: **Performance on New Procgen Tasks** comparing (1) our multi-trajectory transformer conditioned on different number of demonstrations from the same level, (2) Hashmap baseline conditioned on the same demonstrations, and (3) BC baseline conditioned on zero and one demonstration due to context length constraints.

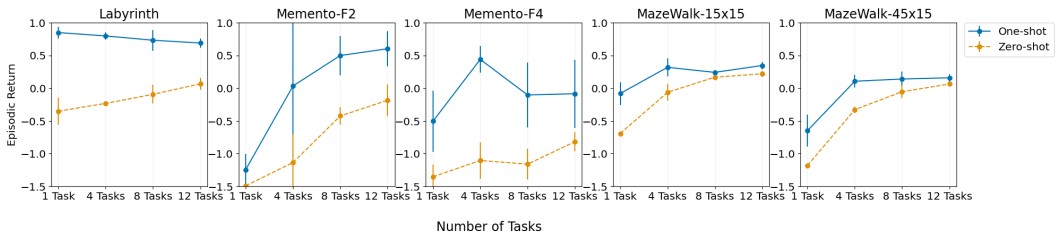

Figure 14: **Effect of Task Diversity**: This figure illustrates the effect of the number of different training tasks on the average episode return for 5 unseen MiniHack tasks. The dashed line indicates the zero-shot performance, and the solid line represents the one-shot performance. Overall, both the one-shot and zero-shot performances improve when increasing the number of tasks from 1 to 8, but then they plateau. This suggests that increasing the task diversity can help both the in-weights and in-context learning but the improvements have diminishing returns and may depend on the similarity between the train and test tasks. The error bars represent the standard deviation across 3 model seeds.

plunder and acts as an oracle in case of Maze. We hypothesize that the hashmap in the case of Maze acts as an oracle since it is very difficult to go out of distribution.

## C.4 TASK DIVERSITY RESULTS

In this section, we extend the results presented in Section 5.2 by showing the performance on unseen environments for a model pre-trained on 4 tasks. We can see that the claims made in Section 5.2 still hold - increasing the diversity of tasks improves generalization to new yet similar tasks via in-weights learning, but it is not clear how the diversity affects the model's ability to 'soft-copy' via in-context learning. In Figure 14 we see that above a certain diversity of tasks, the in-context learning ability plateaus, while for lower task diversity, some novel tasks are capable of demonstrating in-context learning, while others are not.

## D BROADER IMPACT

The ability to generalize to unseen tasks with the help of single demonstration is crucial for many real world applications like robotics, self-driving cars. In this work we focus on how to achieve this goal with the help of in-context learning. More specifically, our work focuses on highlighting how different factors like trajectory burstiness $p_b$, environment dynamics, task dynamics, model and dataset sizes influence in-context learning in sequential decision-making settings. Since our results are based on Minihack and Procgen environments, which are somewhat simplified comapared to real-world settings, we do not foresee any potential negative impact on the society. We believe that our work could provide some useful insights into using transformers for sequential decision-making settings.

## E HYPERPARAMETERS

We present the hyperparameters used in this work in Table 4

