# OpenReview forum: "Learning to Solve New sequential decision-making Tasks with In-Context Learning"
_ICLR.cc/2024/Conference — Submitted to ICLR 2024_

### Official Review · Reviewer_CT2T · 2023-10-31

**Soundness:** 3 good
**Presentation:** 3 good
**Contribution:** 2 fair
**Rating:** 3
**Confidence:** 3

**Summary:**

The paper studies the problem of in-context learning in sequential decision-making settings. The paper finds that it is important for the context to contain full trajectories to cover potential situations at deployment time. The authors provide experiments on MiniHack and Procgen benchmarks, showing the method can generalize to new tasks with just a few expert demonstrations, without weight updates. The work claims to be the first to demonstrate that transformers can generalize to entirely new tasks in these benchmarks using in-context learning.

**Strengths:**

- The paper is well written and easy to follow.
- The paper shows some nice experiments on the MiniHack and Procgen environments, showing how in-context learning can perform well on unseen tasks.
- The paper's setting of in-context learning in decision-making problems is an interesting problem to study.

**Weaknesses:**

- The paper highlights the ability to perform well on unseen tasks but this actually relies on having a lot of demos from related tasks. Can the authors better clarify the relationship between the data they train on and the unseen tasks they evaluate on?
- Novelty-wise, the method is very similar to works like Prompt-DT, except actually requires stronger data assumptions (full expert demos).
- A lot of the insights in the empirical study are not that interesting, e.g. the paper highlights results like showing that in-context learning improves with trajectory burstiness, but it is not surprising that having demos similar to the query inside the context improves the performance. Can the authors give more clarity on what the most surprising, interesting takeaways are from the study?

**Questions:**

See weaknesses above.

---

> ### Author Response · Authors · 2023-11-16
>
> We thank the reviewer for the comments. In this response we will respond to all the comments and questions the reviewer raised.
> 1. **Relying on a lot of expert demonstrations** - Note that we use only 7 expert demonstrations for each level at test time which we’d argue is not a large number for learning (as echoed by reviewer qktf) entirely new tasks with unseen states, actions, dynamics, and rewards.
> 2. **Relationship between the train data and test data** - We ask the reviewer to look at the common response where we clearly explained the difference between the train and test tasks.
> 3. **Comparison with Prompt DT** -  Please refer to our common response where we explain in greater detail about this.
> 4. **Surprising aspect of our work** - Our paper is **first to demonstrate few-shot learning of new Procgen and MiniHack tasks from only a handful of demonstrations, without any online feedback, after training on a relatively small number of other tasks from the same domain (11 in the case of procgen and 12 in the case on MH)**.
> We believe this result is by no means trivial (and it surprised us) given that prior work was either only able to generalize to minor task variations (slightly different reward functions but same states, actions and dynamics, as in PromptDT) or required orders of magnitude more compute and training environments to generalize to more challenging tasks (as in Ada which trains for 100B timesteps on a task pool size of 25B, however at that point it’s not clear test tasks are OOD from train tasks, whereas in our case they are clearly OOD). Note that the train and test tasks are vastly different having different states, actions, dynamics, and reward functions. While we agree the approach is simple we believe this is in fact a strength of our work as it should be easy to adapt, apply to other settings, scale, and build upon by the community.
> Our paper performs an extensive analysis of different design choices and implementation details demonstrating their importance in getting the method to work. We believe these insights will be of interest to researchers and practitioners alike. In particular, we demonstrate a minimum model size, burstiness, data size and diversity are needed to obtain strong results.
> In addition, section 5.2 discusses in depth the limitations of this approach and the fact that it doesn’t work as well in certain settings with high stochasticity or high-stakes scenarios where a single wrong action can be fatal. Again, this supports the point that it is by no means “a given” that this approach works. In fact we believe there is a need for more research in this direction to develop methods that are robust to highly stochastic and unforgiving environments. It’s possible that offline learning and transformers are not even the best solution to these settings. All of this is to say that the results are much more nuanced and we try to convey this in our paper.

---

> > ### Author Response · Authors · 2023-11-20
> > **Any more concerns?**
> >
> > Dear Reviewer,
> >
> > We appreciate the time you have dedicated to reviewing our paper. In the response above and in our common response, we have addressed your concerns regarding the novelty of our work, expert demonstrations, and surprising aspects of our work. If you have no outstanding concerns, we hope you will consider raising your score. Otherwise, please let us know what is standing in the way of recommending acceptance for our paper.

---

### Official Review · Reviewer_F4ph · 2023-10-31

**Soundness:** 2 fair
**Presentation:** 3 good
**Contribution:** 1 poor
**Rating:** 3
**Confidence:** 3

**Summary:**

This work targets the setting of zero-shot and few-shot learning, where the train and test MDPs contain completely separate games (tasks) - in contrast to previous works where the train and test sets contain the same game but with different levels.  The paper proposes to adapt the causal transformer model to this few-shot setting by first training expert agents on each of the training tasks, and then collecting a dataset from the experts’ trajectories to train the transformer model. The authors propose to train the transformer using multi-trajectory sequences rather than single-trajectory sequences and to construct the multi-trajectory training so that the context contains at least one trajectory from the same level as the query. At test time, for the few-shot setting, the transformer is conditioned on 1-7 full expert trajectories, while for the zero-shot setting the transformer is not conditioned on any expert trajectories.

The authors compare their results to two baselines: BC and hashmap, and performed an extensive ablation study containing the dataset sizes, task diversity, environment stochasticity, and trajectory burstiness.

**Strengths:**

* The paper suggests a new setting that has not been studied before - to test on games withheld during training by utilizing expert policies that were trained on the training set (a separate set of games).

* The results show a clear advantage to the proposed approach over the baselines and the authors performed an extensive ablation study.

**Weaknesses:**

* The proposed approach seems as a small adaptation of pre-existing approaches, i.e. causal transformer with multi-trajectory training, to new benchmarks (MiniHack and Procgen) in the offline setting.

* There is no comparison to other offline methods such as CQL [1]

* It is mentioned in the paper that all the results are produced using 3 seeds. In my opinion, for such noisy benchmarks evaluating on only 3 seeds is not enough to reliably estimate the mean and variance.

* The results are not clear to me - for example, in Figure 4 the episodic return is very low compared to the score reported by [2].

**Questions:**

I would like to ask the author to address the following questions:

1. For the few-shot evaluation (when testing the model): is the expert policy, which creates the few trajectories (1-7) for conditioning the transformer, trained on the test games or the training games?
2. Are the above few trajectories (1-7) sampled from the same level as the query level?
3. Why is the return in Figure 4 so low compared to the return reported in [2]?
4. Is the Procgen dataset evaluated on the easy or hard difficulty mode?
5. Are the results in Figure 3 normalized?


A technical detail:
* In the first paragraph of the background - /mu the initial state distribution is not defined.





[1] Kumar, Aviral, et al. "Conservative q-learning for offline reinforcement learning." Advances in Neural Information Processing Systems (2020): 1179-1191.

[2] Cobbe, Karl, et al. "Leveraging procedural generation to benchmark reinforcement learning." International conference on machine learning. PMLR, 2020

---

> ### Author Response · Authors · 2023-11-16
>
> We thank the reviewer for their time. Here are the responses to the questions -
>
> 1. **Novelty** - We wrote a detailed comment about the novelty aspect of our work and how it is different from previous works. Could you please point us to the works, apart from the previous works considered in the common response, which are seemingly similar to the problem setting we consider?
> 2. **Comparison with CQL** - We refer the reviewer to look at the common response where we do comparison of our method with all the related previous works including CQL.
> 3. **Number of seeds** - We also ran 5 seed experiments for procgen experiments and we didn’t notice any big differences in the conclusions. We will update the appendix with 5 seed results.
> 4. **Comparing with Procgen[2]**  -  The results in the original Procgen paper are for a completely different setting than ours so this is not a fair comparison. **In the Procgen paper, agents are trained with PPO on multiple levels of a single game (e.g. Ninja) and tested on unseen levels from the same game (i.e. also Ninja), thus probing IID generalization In our paper, agents are trained offline on data from multiple tasks (e.g. Ninja) and tested on entirely new tasks at test time (e.g. Jumper) thus probing OOD generalization which is more challenging. ** Thus, the lower performance in our paper is expected.
> 5. **Clarity about test demonstrations**  -  The expert demonstrations used at test time come from an expert policy trained on that particular environment. For example, when evaluating our transformer model on Ninja task, we use an expert PPO policy trained only on Ninja to collect expert trajectories and condition on those demonstrations.
> 6. **Few-shot trajectories sampled from the same level?** Yes, the trajectories are from the same level we are testing on. Note that, this is a challenging setting (look at common response for more details) where test tasks are OOD. Moreover, the environments are stochastic which adds more challenges. Hence the model might still struggle to generalize to OOD tasks just from a few demonstrations.  We will clarify this in the paper.
> 7. **Procgen task difficulty** - We consider “easy” difficulty mode in our experiments.
> 8. **Normalized scores** - No, we do not normalize anything and we report the raw scores for both MiniHack and Procgen.

---

> > ### Author Response · Authors · 2023-11-20
> > **Any more concerns?**
> >
> > Dear Reviewer,
> >
> > We appreciate the time you have dedicated to reviewing our paper. In the response above and in our common response, we have addressed your concerns regarding the novelty of our work, the evaluation protocol, and other aspects. If you have no outstanding concerns, we hope you will consider raising your score. Otherwise, please let us know what is standing in the way of recommending acceptance for our paper.

---

> > > ### Comment · Reviewer_F4ph · 2023-11-22
> > >
> > > I thank the authors for their answers and clarifications. Some of my concerns were clearly addressed. However, I still have reservations about the quality of the evaluations and the presented results. Especially regarding limited baseline comparisons.
> > > Although the setting presented in the paper is indeed different from the setting that existing works (Prompt DT, CQL, Algorithmic Distillation, etc.) are targeting and were designed for, the proposed algorithm relies heavily on offline datasets (sampled from experts that were trained on different games). Therefore, comparing the proposed method to the performance of a leading offline baseline (trained in the offline setting, only on the training games) would highlight the performance benefit of the proposed few-shot adaptation phase. This will contribute to the understanding of the importance of the different components of the algorithms.
> > >
> > > In addition, training on only 3 seeds is not enough to draw definitive conclusions. I would urge the author to show results trained on more seeds (or at least a subset of the results).
> > > For these reasons, I’ll keep my score unchanged for now.

---

### Official Review · Reviewer_phhR · 2023-11-01

**Soundness:** 3 good
**Presentation:** 2 fair
**Contribution:** 2 fair
**Rating:** 3
**Confidence:** 4

**Summary:**

This work studies the use of transformers for generalization to new tasks from MiniHack and Procgen. Their experiments show that a model pre-trained on different levels and tasks of Procgen can learn a new task from a few demonstrations of the task. The transformer is trained with demonstration contexts that can either be from the same or different levels as the query. There is also additional empirical analysis on different variables, such as whether the context is from the same level, environment stochasticity, and task diversity.

**Strengths:**

- This work shows that transformers can perform few-shot imitation learning on new Procgen tasks, which has not been explicitly shown previously.

- The paper is written pretty well at a low-level and tries to be thorough in its experiments through additional experiments to understand failure modes and effect of different environmental and algorithmic factors.

- Given the significance of in-context learning in large language models, it seems timely and appropriate to study it in the context of decision-making.

- The environment stochasticity result is pretty interesting, i.e., that the model can learn copying behavior if the training environments are deterministic.

**Weaknesses:**

- Existing papers such as Prompt-DT (Xu et al, 2022) and AdA (Team et al, 2023) have shown similar results (in some cases, with even less presumptive data than full demonstrations) though in different domains. So it's perhaps not too surprising that we see this type of generalization in Procgen as a result. The main result that models trained with demonstrations in the context can perform better than models without the demo context is also expected.

- There are a couple of claims that do not seem sufficiently supported: (1) meta-RL methods "tend to be difficult to use in practice and require more than a handful of demonstrations or extensive fine-tuning," (2) "in sequential decision-making it is crucial for the context to contain full trajectories (or sequences of predictions) to cover the potentially wide range of states the agent may find itself in at deployment" (see Questions).

- The concepts of burstiness from Chan et al and trajectory burstiness have a pretty weak relation. In the case of this paper, it seems pretty clear from the get-go that demonstration contexts from the same level as the query would be more relevant than from any other levels.

**Questions:**

- What does trajectory burstiness mean for the zero-shot model in Fig. 6(a)?

- "[Meta-RL methods] tend to be difficult to use in practice and require more than a handful of demonstrations or extensive fine-tuning" --> Including some of these comparisons in the experiments would be help support this statement.

- "Our key finding is that in contrast to (self-)supervised learning where the context can simply contain a few different examples (or predictions), in sequential decision-making it is crucial for the context to contain full trajectories (or sequences of predictions)
to cover the potentially wide range of states the agent may find itself in at deployment." --> Full trajectories as opposed to what? The experiments only show comparisons between full demos vs no demos, and didn't study other potential contexts, such as partial demos or non-expert trajectories. I think a study of different potential contexts and what is required for in-context learning would be interesting.

- "This means that the agent manages to perform well on the new task even without copying actions from its context. This suggests the model is leveraging information stored in its weights during training, also referred to as in-weights learning" --> Could you elaborate on how not copying the context actions suggests in-weights learning as opposed to ICL? This conclusion seems to equate ICL with the ability to copy context actions.

---

> ### Author Response · Authors · 2023-11-16
>
> We thank the reviewer for thoughtful comments. We appreciate the reviewers comments on strengths of our work. In this response, we will address the questions you raised in your review.
>
> 1. **PrompDT and AdA**: We would like the reviewer to refer to the common response where we contrast our method with prior work. We differ from PromptDT paper in terms of many aspects including multiple environment training, scale of our models, OOD evaluations etc. With regards to AdA, we would like to highlight that the setting is different from our work. AdA considers online RL setting with transformers in contrast with offline few-shot imitation learning setting, and requires orders of magnitude more compute and training environments to generalize to more challenging tasks (100B timesteps on a task pool size of 25B). Regarding surprising aspects of the results, we want to emphasize again that the setting we consider in this paper is very challenging and to the best of our knowledge this setting hasn’t been explored in prior work.
> 2. **Meta RL** - The problem setting considered in our paper differs from that of Meta-RL, where training is online or offline and, at test time, the agent learns from online interactions and feedback (i.e., rewards) within a new environment. In contrast, our paper focuses on the few-shot imitation learning setting, where training occurs offline based on expert demonstrations from a set of tasks, and at test time, the agent learns a new task from only a handful of offline demonstrations without interactions or feedback in the new environment. Therefore, while Meta-RL is loosely related – a fact we acknowledge in the related work section – these approaches are not directly comparable, as they operate under vastly different assumptions. However, following your suggestion, we will rephrase that sentence to ensure our claims are well supported by our experiments.
> 3. **Copying Actions** - Thank you for pointing this out. Recent works in ICL, such as 'Incontext Learning and Induction Heads'[1] by Anthropic, suggest that 'induction heads' are a key feature in transformer models for ICL. Induction head is  a circuit whose function is to look back over the sequence for previous instances of the current token (call it A ), find the token that came after it last time (call it B ), and then predict that the same completion will occur again (e.g. forming the sequence [A][B] … [A] → [B] ). This is what we mean by copying in our context. We understand that the term 'copying' is used loosely in the paper which caused this confusion. We will rephrase our sentence structure to reflect this understanding, including relevant citations.
> 4. **Partial Demonstrations** - We want to highlight that we consider partial demonstrations in our procgen experiments because of the long episode lengths in some of the procgen tasks.
> 5. **What does burstiness 0.0 mean?** Burstiness 0.0 means that in the entire dataset, there is a zero probability of finding the sequences which are bursty.
> 6. **Comparison to burstiness in Chan et al.** We think the burstiness in chan et al and trajectory burstiness are closely connected. Chan et al defines a bursty sequence as a sequence where there are multiple occurrences of the same classes which are similar to query (Figure 1 in [1]). Similarly, in our work, a trajectory bursty sequence is a sequence where there are multiple trajectories which are from the same level as the query.

---

> > ### Author Response · Authors · 2023-11-20
> > **Any more concerns?**
> >
> > Dear Reviewer,
> >
> > We appreciate the time you have dedicated to reviewing our paper. In the response above and in our common response, we have addressed your concerns regarding the novelty of our work (specifically wit PromptDT, AdA and MetaRL), the trajectory burstiness, and other aspects. If you have no outstanding concerns, we hope you will consider raising your score. Otherwise, please let us know what is standing in the way of recommending acceptance for our paper.

---

### Official Review · Reviewer_qktf · 2023-11-06

**Soundness:** 3 good
**Presentation:** 2 fair
**Contribution:** 2 fair
**Rating:** 5
**Confidence:** 3

**Summary:**

This paper studies the problem of in-context learning for decision-making. A method of training transformers is proposed where expert demonstrations are generated across many tasks and the model is expected to predict expert behavior from this context. The method is demonstrated on procgen and nethack, two challenging RL settings.  It is shown that in-context learning can be achieved from a handful of demonstrations in order to generalize to new test tasks.

**Strengths:**

The problem is important and interesting: the in-context abilities of transformers for decision-making problems is comparatively understudied relative to supervised learning problems. This paper contributes to a growing understanding of decision-making with transformers.

The generalization of the method to entirely new tasks in procgen and nethack is impressive as these are challenging settings and each task is quite different from the others. There are really only a handful of training tasks.

The analytical studies are thorough and mostly informative, especially the one showing how the performance varies with the number of training tasks and the one on failure modes.

**Weaknesses:**

Overall, I think this is a good paper with a thorough analysis, but there are two main weaknesses of the paper: clarity and novelty/significance.

Clarity: both the problem setting and the methodology of the training are not very clear and this makes it difficult to understand the significance of the results.

- During testing, the agent is given a handful of expert demonstrations. Are these all demonstrations on the same task and same level? If not, how does this work for the baselines hashmap if they are using demos from different levels? If so, why is the transformers trained with several sequences of demos from different levels? Why not just train with demonstrations from the same level and task always?
- Related to this, how do I interpret this, which suggests that all demos in the context come from the same level: “we collect offline data from 11 Procgen tasks and train a transformer on Procgen sequences compromising of five episodes from the same level.” What does burstiness even mean here if the levels are never varied?
- What does it mean for BC-1 to condition on ‘one demonstration’? Does this mean you give it full demonstration in the same task and same level? In other words, does the context look like this: [expert demo, history observed so far]. How would this be different from your method if you were just limited to training on just two sequences?
- What is the maximal achievable reward in each of the environments? This could be helpful to better understand the final results.

It would further be helpful to distinguish the work from prior methods better. A more thorough comparison would help readers with a better understanding of the present problem setting and method.

- The method appears to be very similar to Prompt-DT [1] perhaps without the return conditioning. There’s already a short discussion in the related but this ought to be carefully dissected, I think.
- The method is also very similar to DPT [2], which also considered training and conditioning on expert demonstrations to solve new tasks. If there is a difference, both of these papers seem like highly relevant baselines.
- It is also likely worth distinguishing the method with other in-context RL works like [3] and [4].

Beyond transformers, there’s additional work on meta/few-shot imitation learning that could be helpful to discuss.

As a result, the overall takeaways are a bit hard to discern. It’s clear now that there are multiple solid contributions in this paper (a method and a thorough analysis), but I think the takeaways could be better communicated.

[1] Xu M et al. Prompting decision transformer for few-shot policy generalization. International conference on machine learning 2022.

[2] Lee JN et al. Supervised Pretraining Can Learn In-Context Reinforcement Learning. arXiv preprint arXiv:2306.14892. 2023.

[3] Lu C et al. Structured state space models for in-context reinforcement learning. arXiv preprint arXiv:2303.03982. 2023.

[4] Laskin M et al. In-context reinforcement learning with algorithm distillation. arXiv preprint arXiv:2210.14215. 2022.

**Questions:**

See above section for specific questions. Misc:

- How long are each of these sequences? I.e. what is T?
- In what settings would you expect this method would work (or these analysis be useful) under the current assumptions, beyond gameplaying?

---

> ### Author Response · Authors · 2023-11-16
> **Official Response [Part 1/2]**
>
> We thank the reviewer for their thoughtful comments and feedback. We appreciate the reviewer acknowledging the challenges posed by our setting, our thorough experimental analysis and discussion on limitations of our work.  In this response, we hope to successfully answer your remaining concerns.
>
> 1. **Similarities with PromptDT** - We wrote a detailed response to this concern in the “common response” section above where we contrast our work with  all the papers mentioned in your review. We encourage you to read the response and let us know if you have any additional questions.
> 2. **Additional discussion on Related works** - We agree that additional discussion about recent works would be beneficial. We will add the works you mentioned in your review and extend our related works section in the appendix.
> 3. Regarding Evaluation
>     * **Few-shot evaluation** - During few-shot evaluation, the model is given access to a limited number of expert demonstrations from the task on which it is being evaluated. For instance, when assessing performance on a Ninja task from Procgen, we condition our model on a few expert demonstrations of the Ninja task, specifically from the same level on which it will be tested. We then unroll the policy of the transformer in the online environment and measure the obtained reward. This itself is a challenging evaluation because the model never saw the “Ninja” task during the training, hence OOD, and learning from just a handful of demonstrations in this scenario could be very challenging.
>     * **Hashmap evaluation** - Hashmap baseline evaluation is similar to that of the model. Again, taking the example of the “Ninja” task, we hash a few expert demonstrations coming from the same level on which it will be tested. This baseline suffers if the environment is stochastic but could act as an oracle in case the environment is deterministic. Finally, we use the same demonstrations for all the baselines and our method for fair comparison.
>     * **Training** - When the trajectory burstiness probability is 1.0 (which is the case for figure 3 and 4), in each sequence the demonstrations come from the same level of a given task. So each sequence in the training set looks like this [ demonstration 1 from task A level A, demonstration 2 from task A level A, demonstration 3 from task A level A, demonstration 4 from task A level A ….]. In this work, we have 10k levels per  task and 11 training tasks in case of procgen resulting in 11 * 10k = 110K training sequences.
> 4. **Sequence construction** - If all or at least two demonstrations are coming from the same level in a given sequence, then the sequence is called a bursty sequence with trajectory burstiness probability 1.0. This means that the relevant information required to solve the task is entirely in the sequence hence this design aids in-context learning. However, note that the trajectories, despite being from the same level, may vary because of the inherent environment stochasticity which means that the model still needs to generalize to some extent  to handle these cases.
> 5. **Behavioral Cloning (BC-1) baseline** - Yes, BC-1 means during the *evaluation time* (and *not* during the training), we condition the model with one expert demonstration. This is different from our method because during training BC-1 and BC-0 are trained with a single trajectory sequence and they only differ just in the way they are evaluated. The sequence construction looks exactly like you rightly mentioned. The aim of BC-1 baseline is to check if the model is ever making use of the expert demonstration info during the test time. However, this evaluation could be somewhat OOD for the model to generalize which is reflected in the results.
> 6. **Upper bound of performance** - We will add these numbers in the appendix. However, we do want to highlight that, to our knowledge, there are not so many works which consider the setting we consider in our paper so it is hard to benchmark our current results. Ideally we want to move our numbers close to that of an oracle which is obtained by doing online RL from scratch in the test environment and these are already reported in [1].

---

> > ### Author Response · Authors · 2023-11-16
> > **Official Response [Part 2/2]**
> >
> > 7.  **Value of T** -  In case of Minihack, the average episode length  (T) is about 100 steps. Incase of procgen, some tasks have very large episodes (> 1000 steps). So, to accommodate multiple trajectories, we truncate the episodes to have at maximum of 200 steps.  For the procgen evaluations, we do transformer rollouts for a maximum of 200 steps for that reason.
> > 8. **Applicability beyond game playing** - We would like to highlight that our approach doesn’t use any domain-specific information or assumptions, making it broadly applicable for any other domains beyond games. We demonstrate that our approach works for two different domains without any specialized adjustments (Procgen and MiniHack), further supporting the generality of our method. Note that these domains have vastly different observation spaces, one consisting of discrete grids and the other high-dimensional images.
> > One limitation of our approach, which we discuss in section 5.2, is the finite context length so if the trajectories are longer than the context our method suffers. Also, we expect performance to be worse in very stochastic or "unforgiving" environments where a single wrong action is fatal. Please see Section 5.2 where we talk extensively about these failure modes and show how they affect performance.
> >
> > [1] Cobbe, Karl, et al. "Leveraging procedural generation to benchmark reinforcement learning." International conference on machine learning. PMLR, 2020

---

> > > ### Author Response · Authors · 2023-11-20
> > > **Any more concerns?**
> > >
> > > Dear Reviewer,
> > >
> > > We appreciate the time you have dedicated to reviewing our paper. In the response above and in our common response, we have addressed your concerns regarding the novelty of our work, the evaluation protocol, baselines, and other aspects. If you have no outstanding concerns, we hope you will consider raising your score. Otherwise, please let us know what is standing in the way of recommending acceptance for our paper.

---

### Author Response · Authors · 2023-11-16
**Common Response Pae**

## How are the contributions in this paper different from previous works? -

In this response, we aim to clarify some common misunderstandings regarding our paper’s core contributions and how they differ from prior work.



**Setting** - We wish to emphasize the complexity of the setting we consider in this paper. In contrast to most prior research that reviewers have referenced, our work tackles generalization to **not only unseen but also out-of-distribution (OOD) environments with different states, actions, dynamics, and reward functions**. As defined in [5], OOD generalization pertains to conditions where the training and testing distributions do not overlap (refer to Figure 1 in [5]). For instance, as detailed in Table 3 of our paper, the training tasks for procgen experiments include 11 Procgen tasks (Bigfish, Bossfight, Caveflyer, Chaser, Fruitbot, Dodgeball, Heist, Coinrun, Leaper, Miner, Starpilot), while the remaining 5 tasks (Climber, Ninja, Plunder, Jumper, Maze) are reserved for testing. For example, in the case of Bigfish, which is in the training set in our case, the agent must only eat fish smaller than itself to survive. It receives a small reward for eating a smaller fish and a large reward for becoming bigger than all other fish. In the test set, one of the environments we have is Ninja, where the agent jumps across ledges avoiding bombs, charging jumps over time, and can clear bombs by tossing throwing stars. The agent is rewarded for collecting the mushroom that ends the level. As one can see, the train and test tasks are completely different and it is hard to solve the test tasks without having any prior knowledge about it. To better grasp how different these tasks are, we encourage  the reviewers to look at videos of agents playing all Procgen games: https://openai.com/research/procgen-benchmark

This difference in the problem setting  distinguishes our work from the existing literature. To the best of our knowledge, our work is first to demonstrate competitive results in this challenging setting.
In the table below, we further clarify the distinctions between our method and prior works. We will include this table in the camera ready version if the paper gets accepted.




| Paper | Setting | Multi-Environment Training | Train Environments  | Test Environments | Type of Generalization according to [5] | Axes of Generalization |
|------|------|------|------|------|------|------|
| Our Work  |Few-Shot Imitation Learning   | Yes     |  11 Procgen tasks (Bigfish, Bossfight, Caveflyer, Chaser,Fruitbot, Dodgeball, Heist, Coinrun, Leaper, Miner, Starpilot) Note - We also have similar experiments on MiniHack |  Remaining 5 procgen tasks (Climber, Ninja, Plunder, Jumper,Maze) and four MiniHack tasks.  | OOD generalization with disjoint train and test environments     |   States, rewards and dynamics |
| Prompt DT [1]  |  Offline RL as a Sequence Modeling problem with transformers  |  No    |  Cheetah vel, Cheetah-dir, ant-dir,Meta-World reach-v2, Dial  | Same environments with different reward function   |  IID generalization   |  Rewards    |
|  CQL [2]    |  Offline RL  | No | Gym domains, Adroit, Atari, AntMaze |  Same as train    |   NA   |  NA  |
| Algorithmic Distillation [3]   | Offline RL as a Sequence Modeling problem with transformers  | No | Dark Room, Dark-Key-Door, DMLab Water Maze  | Unseen goals of the same train environments   | IID generalization | Rewards |
|  Structured state space sequence Models [4]    |  Online RL with Structured State Space Models    |   Yes   |   Ball-in-cup catch, finger-turn hand, cartpole swingup, Pointmass easy, Finger spin, Reacher easy   | Hard versions of the same train tasks     | OOD Generalization     | Rewards and Dynamics but not States |
|   DPT [6]    |   Offline RL as a Sequence Modeling problem with transformers    | No     |   Miniworld and Dark Room   |  Unseen goals in dark room and miniworld    |  IID Generalization    | Rewards |
|   AdA[7]    |   Online RL with Transformers   | Yes     |   XLand  |  Unseen tasks in XLand    |  OOD Generalization    | Rewards, states, dynamics |

---

> ### Author Response · Authors · 2023-11-16
> **Common Response [Part 2/2]**
>
> ## Comparison with PromptDT -
> Here, we point out the core differences between PromptDT and our work.
>
> | Our Method  | PromptDT |
> |-----|------|
> |Multiple and diverse training tasks | Single training task|
> |Generalization to new tasks with different states, actions, dynamics, and reward functions. | Generalization to a task with the same states, actions, dynamics, but different reward functions. |
> | Sequence has N trajectories | Sequence consists of prompt trajectories and a query trajectory. |
> | Loss is applied on all trajectories| Loss is applied only on the last trajectory |
> | Long context windows: 2048 tokens | Uses very small context windows: k=20 |
> | Uses expert demonstrations | Use both expert and non-expert demonstrations. |
> | Our architecture is scalable GPT-2 style architecture which has proven to be effective in language modeling literature. We train comparatively larger models with the largest being 300M. We also demonstrate scaling trends with respect to the number of parameters and we show that these models generalize only at scale. | The model architecture the authors use here consists of one attention head, 3 layers with 128 embedding dimensions. Moreover, the model sizes are very small with the largest being under 5M. PromptDT does not provide any scaling analysis. |
>
> Our work has some parallels with PromptDT in how we construct sequences. Specifically, the sequences in the PromptDT paper are structured as [(prompt_trajectories), query trajectory], with the transformer being optimized using MSE loss solely on the query action predictions. Early in our research, we conducted experiments where the loss was exclusively focused on the query predictions and observed that the models had difficulties in generalizing at scale. We discovered that the design choices we emphasize in our paper like trajectory burstiness, data and model scale and environment stochasticity are essential for enabling models to few-shot learn new sequential decision-making tasks in entirely new tasks, including unseen states, actions, dynamics, and rewards.
>
> Comment about Partial Trajectories in PromptDT -  Finally we want to highlight that PromptDT evaluated with partial demonstrations (not expert) during the test time and they do show promising results. However, the caveat here is that 1) their models are conditioned on return-to-go tokens and 2) their experiments are continuous control tasks. For these continuous control tasks, the task can be defined just with the help of achievable return. This means that they do not need full demonstrations to perform well on new tasks. This is vastly different from our setting and we are not sure if promptDT would be successful in the setting we consider.
>
>
> [1] Xu M et al. Prompting decision transformers for few-shot policy generalization. International conference on machine learning 2022.
>
> [2] Kumar, Aviral, et al. "Conservative q-learning for offline reinforcement learning." Advances in Neural Information Processing Systems (2020): 1179-1191.
>
> [3] Laskin M et al. In-context reinforcement learning with algorithm distillation. arXiv preprint arXiv:2210.14215. 2022.
>
> [4] Lu C et al. Structured state space models for in-context reinforcement learning. arXiv preprint arXiv:2303.03982. 2023.
>
> [5] Kirk R et al. “A Survey of Zero-shot Generalisation in Deep Reinforcement Learning” Journal of Artificial Intelligence Research 76 (2023) 201-264
>
> [6] Lee JN et al. Supervised Pretraining Can Learn In-Context Reinforcement Learning. arXiv preprint arXiv:2306.14892. 2023.
>
> [7] Adaptive Agents Team, “Human-Timescale Adaptation in an Open-Ended Task Space”.

---

### Meta-Review · Area_Chair_amzV · 2023-12-11

**Metareview:**

The paper proposes interesting results for applying in-context learning to sequential decision-making tasks. All reviewers appreciated the positive results, but also demonstrated concerns over novelty/thoroughness of the results/clarity of exposition. In the view of the area chair, this work may have a strong impact if the issues raised by the reviewers can be well-addressed, especially the clarity of the problem setup. Unfortunately, it is still below the bar of acceptance at its current state.

**Justification For Why Not Higher Score:**

As the summary mentioned, there are still concerns around novelty/thoroughness of the results/clarity of exposition and reviewers are still not satisfied with the current version.

**Justification For Why Not Lower Score:**

N/A

---

### Decision · Program_Chairs · 2024-01-16

Reject